# ON FLAT MINIMA, LARGE MARGINS AND GENERALIZABILITY

## ABSTRACT

The intuitive connection to robustness and convincing empirical evidence have made the flatness of the loss surface an attractive measure of generalizability for neural networks. Yet it suffers from various problems such as computational difficulties, reparametrization issues, and a growing concern that it may only be an epiphenomenon of optimization methods. We provide empirical evidence that under the cross-entropy loss once a neural network reaches a non-trivial training error, the flatness correlates (via Pearson Correlation Coefficient) well to the classification margins, which allows us to better reason about the concerns surrounding flatness. Our results lead to the practical recommendation that when assessing generalizability one should consider a margin-based measure instead, as it is computationally more efficient, provides further insight, and is highly correlated to flatness. We also use our insight to replace the misleading folklore that small-batch methods generalize better because they are able to escape sharp minima. Instead we argue that large-batch methods did not have enough time to maximize margins and hence generalize worse.

## 1 INTRODUCTION

Understanding under which conditions a neural network will generalize from seen to unseen data is crucial, as it motivates design choices and principles which can greatly improve performance. Complexity or generalization measures are used to quantify the properties of a neural network which lead to good generalization. Currently however, established complexity measures such as VC-Dimension (Vapnik, 1998) or Rademacher Complexity (Bartlett & Mendelson, 2002) do not correlate with the generalizability of neural networks (e.g. see Zhang et al. (2016)). Hence many recommendations, such as reducing model complexity, early stopping, or adding explicit regularization are also not applicable or necessary anymore. Therefore, there is an ongoing effort to devise new complexity measures that may guide recommendations on how to obtain models that generalize well.

A popular approach is to consider the flatness of the loss surface around a neural network. Hochreiter & Schmidhuber (1997) used the minimum description length (MDL) argument of Hinton & Van Camp (1993) to claim that the flatness of a minimum can also be used as a generalization measure. Motivated by this new measure Hochreiter & Schmidhuber (1997), and more recently Chaudhari et al. (2019), developed algorithms with explicit regularization intended to converge to flat solutions. Keskar et al. (2016) then presented empirical evidence that flatness relates to improved generalizability and used it to explain the behavior of stochastic gradient descent (SGD) with large and small-batch sizes. Other works since have empirically corroborated that flatter minima generalize better (e.g. Jiang et al. (2019); Li et al. (2018); Bosman et al. (2020)).

There are however various issues that are still unresolved, which makes using flatness for constructing practical deep learning recommendations difficult. For one, flatness is computationally expensive to compute. The most common way to compute the flatness is via the Hessian, which grows quadratically in the number of parameters; this becomes too large when used with modern networks containing millions of parameters. It is also not clear to what extent flatness is a true measure of generalizability, capable of discerning which neural network will or will not generalize. Dinh et al. (2017) showed that reparametrizations affect flatness and a flat model can be made arbitrarily sharp without changing any of its generalization properties. In addition Probably Approximately Correct (PAC-Bayes) bounds that bound the generalizability in terms of the flatness are also either affected

by rescaling, impossible to evaluate or loose (Neyshabur et al., 2017; Arora et al., 2018; Petzka et al., 2020). While there have been solutions attempting to prevent issues around reparametrization (Liang et al., 2019; Tsuzuku et al., 2019), it remains to establish whether flatness is an epiphenomenon of stochastic gradient descent or other complexity measures as Achille et al. (2018) and Jastrzebski et al. (2018) are suggesting. This motivates investigating possible correlations to more well-understood measures of generalization that may help alleviate issues surrounding flat minima, while allowing flat minima to be used when appropriate.

In this paper we will demonstrate a correlation to classification margins, which are a well-understood generalization measure. Margins represent the linearized distance to the decision boundaries of the classification region (Elsayed et al., 2018). An immediate consequence of such a relationship is that to assess generalizability, we could now simply use a computationally cheap and more robust margin based complexity measure. Our contributions will demonstrate further practical implications of the relationship between margins and flatness which open doors to valuable future work such as a better understanding of why and when a model generalizes and more principled algorithm design.

- We prove that under certain conditions flatness and margins are strongly correlated. We do so by deriving the Hessian trace for the affine classifier. Based on its form, we derive an expression in terms of classification margins which we show correlates well with the Hessian trace, with increasing training accuracy for various neural network architectures. By being able relate the two complexity measures, we are now able to provide various practical recommendations, and offer different perspectives on phenomena that may not be explainable without such a view. These are shown in the following contributions.

- We use our insight to replace the misleading folklore that, unlike large-batch methods, small-batch methods are able to escape sharp minima (Keskar et al., 2016). We instead employ a margin perspective and use our empirical results along with recent results by Banburski et al. (2019) and Hoffer et al. (2017) to argue that a large batch method was unable to train long enough to maximize the margins. With our explanation, we help reframe the small and large-batch discussion and build further intuition.

- We show that once a neural network is able to correctly predict the label of every element in the training set it can be made arbitrarily flat by scaling the last layer. We are motivated by the relationship to margins which suffer from the same issue. We highlight this scaling issue because, in some instances, it may still be beneficial for algorithm design to be guided by convergence to flat regions. Hence, we need to account for scaling issues which make it difficult to use flatness to assess whether a network generalizes better than another.

Other works have made connections between flatness and well-behaved classification margins via visualizations (see Huang et al. (2019); Wang et al. (2018)), but they have not demonstrated a quantifiable relationship. Further work has used both the classification margins and flatness to construct PAC-Bayes bounds (Neyshabur et al., 2017; Arora et al., 2018), and have related flatness to increased robustness (Petzka et al., 2020; Borovykh et al., 2019) however they did not show when and to what extent these quantities are related.

We structure the paper as follows. In Section 2, we discuss both our notation and our motivation choosing the cross-entropy loss and the Hessian trace as the flatness measure and provide further background on the classification margins. In Section 3, we present our contribution showing a strong correlation between the margins and flatness by deriving. In Section 4, we combine recent results based on classification margins to offer a different perspective on the misleading folklore on why larger-batch methods generalize worse. In Section 5, we highlight that networks can be made arbitrarily flat. Lastly, we offer our thoughts and future work in the Section 6.

## 2 PROBLEM SETTING

We first define the basic notation that we use for a classification task. We let $\mathcal{X}$ represent the input space and $\mathcal{Y} = \{1, ..., C\}$ the output space where $C$ are the number of possible classes. The network architecture is given by $\phi : \Theta \times \mathcal{X} \to \mathbb{R}^{|\mathcal{Y}|}$ where $\Theta$ is the corresponding parameter space. We measure the performance of a parameter vector by defining some loss function $\ell : \mathbb{R}^C \times \mathcal{Y} \to \mathbb{R}$. If we have have a joint probability distribution $\mathcal{D}$ relating input and output space then we would

like to minimize the expected loss $\mathcal{L}_{\mathcal{D}}(\theta) = \mathbb{E}_{(x,y)\sim\mathcal{D}}[\ell(\phi(\theta,x),y)]$. Since we usually only have access to some finite dataset $D$, we denote the empirical loss by $\tilde{\mathcal{L}}_D(\theta) = \frac{1}{|D|}\sum_{i=1}^{|D|}\ell(\phi(\theta,x_i),y_i)$. If $\mathcal{L}_{\mathcal{D}}$ and $\tilde{\mathcal{L}}_D$ are close, then we would say a model generalizes well, as we were able to train on a finite dataset and extrapolate to the true distribution. We will use the cross-entropy loss which is given by $\ell(\phi(\theta,x),y) = -\log(S_y(\phi(\theta,x)))$ where the softmax function $S : \mathbb{R}^C \to \mathbb{R}^C$ is given by $S(a)_i = \frac{e^{a_i}}{\sum_{j=1}^C e^{a_j}}$ (see Goodfellow et al. (2016)).

The choice of the cross-entropy function as the loss function has a significant impact on how the flatness measure behaves. Unlike the multiclass mean squared error (MMSE), exponential type losses such as the cross-entropy loss on neural networks have been shown to include implicit regularization which leads to margin maximizing solutions for neural networks (Banburski et al., 2019). Also, various properties for flat minima which have been proven for the MMSE loss by Mulayoff & Michaeli are not applicable to the cross-entropy loss, further highlighting the fundamental differences between the loss functions. While the MMSE loss has shown some promise for many classification tasks (Hui & Belkin, 2020) the cross-entropy loss is still the loss which is most used and was primarily used for the empirical evidence around flat minima (Keskar et al., 2016; Chaudhari et al., 2019), which motivates our choice.

The qualitative description of a flat region was given by Hochreiter & Schmidhuber (1997) as "a large connected region in parameter space where the error remains approximately constant". We measure the flatness by the trace of the Hessian of the loss with respect to the parameters (in short the Hessian trace) denoted by $Tr(\mathbf{H}_\theta(\tilde{\mathcal{L}}_D(\theta))$ (Dinh et al., 2017). Since the Hessian is symmetric, the Hessian trace is equivalent to the sum of its eigenvalues which for a fixed parameter space is proportional to the expected increase of the second order approximation of the loss around a fixed minimum $\theta$ in a random direction $\theta'$ with $\theta' \sim \mathcal{N}(\theta,I)$. Since we apply flatness arguments only close to minima, we assume that all eigenvalues are positive and that the Hessian trace is a good measure of flatness Sagun et al. (2017). Even though the Hessian is only an approximation of flatness, the Hessian is often preferred as it allows us to reason about various directions in parameter space via its eigenvectors and eigenvalues (see Sagun et al. (2017); Chaudhari et al. (2019)) and alleviates the issue of infinitely long but sharp ridges making a minimum infinitely flat (Dinh et al., 2017; Freeman & Bruna, 2016). The Hessian has also been linked to feature robustness via its use in the second order approximation of the loss (e.g. Petzka et al. (2020); Borovykh et al. (2019)) and is a promising quantity to relate to the margins.

As we are working with non-linear functions it is intractable to compute exact distances to the decision boundary, therefore we use a measure which is related to the linearized distance as described in Elsayed et al. (2018). Under this view, larger margins are better because the data is further from the decision boundary. Specifically, we define the margins as in Neyshabur et al. (2017): for some vector $v \in \mathbb{R}^C$ and label $y$ we let the margin of $v$ be $\gamma(v,y) = |v_y - \max_{j\neq y} v_j|$. Since we use the margin in different contexts we define the output margins $\gamma(\phi(\theta,x),y)$ and the margins of the model output after the softmax layer $\gamma(S(\phi(\theta,x)),y)$. Due to the intuition of margins relating to the regularity of the classification regions, they have been proven and shown to be a good generalization measure for linear networks (Langford & Shawe-Taylor, 2003) and later for neural networks (see Bartlett et al. (2017); Jiang et al. (2018; 2019)) when correctly adjusted. Due to results by Banburski et al. (2019) and Soudry et al. (2018), Poggio et al. (2019) claimed that a large part of the mystery around generalizability has been solved, since standard optimization methods are maximizing the margin instead of memorizing data.

## 3 THE MARGIN AND HESSIAN TRACE RELATIONSHIP

### 3.1 THE AFFINE CROSS-ENTROPY HESSIAN TRACE

Generally, it is difficult to derive a closed form solution of the Hessian trace due to the non-linear nature of neural networks. To gain insight into what may determine the flatness or sharpness of a solution we consider an affine prediction function for which we derive the following simple and insightful expression for the Hessian trace:

**Proposition 3.1** (Affine Cross-Entropy Hessian Trace (ACEHT)). *Assume an affine predictor given by $\phi((\theta, b), x) = \theta x + b$ where $(\theta, b) \in \mathbb{R}^{C \times d} \times \mathbb{R}^C = \Theta$. Then the trace of the Hessian under the cross-entropy loss assuming our predictor function is:*

$$Tr(\mathbf{H}(\ell(\phi((\theta, b), x), y))) = (|x|^2 + 1)(1 - \sum_{j=1}^{C} S_j^2(\phi(\Theta, x)))$$

$$= (|x|^2 + 1)(1 - |S(\phi(\Theta, x))|^2).$$

The derivation is in Appendix C. We immediately observe that the trace of the Hessian is a product of both the size of the input and $1 - \eta(S(\phi(\theta, x)))$ where $\eta(S(\phi(\theta, x))) = \sum_{j=1}^{C} S_j^2(\phi(\Theta, x))$, where we can view $1 - \eta(S(\phi(\theta, x)))$ as a confidence measure. In the visualization provided in Figure 1 we clearly see that $1 - \eta(S(\phi(\theta, x)))$ is only zero when the predictor predicts one class with probability 1, regardless of whether it is the correct class or not. When the model is least confident, namely when every entry is predicted with probability $1/C$, then $1 - \eta(S(\phi(\theta, x)))$ is also highest. Hence, in the affine case with a cross-entropy loss the Hessian trace can be seen as an indication of the model confidence in its prediction. This confidence interpretation is also connected to classification margins by observing that $S_y \geq \gamma(S(\phi(\theta, x)), y)$ and hence $(1 - \sum_{j=1}^{C} S_j^2(\phi(\Theta, x)) \leq 1 - S_y^2((\phi(\Theta, x))) \leq 1 - \gamma^2(S(\phi(\theta, x)), y)$. Therefore, if the margins are large then the region will also be flat. The intuition for this is that the error in the upper bound becomes smaller as $S_y$ becomes larger, i.e. when the model predicts correctly and confidently. We will also provide evidence for a converse, i.e. a flat minimum has large margins, in the following experimental sections. Finally, we note that without the expression in Proposition 3.1 we would not have been able to derive the upper bound $1 - \gamma^2(S(\phi(\theta, x)), y)$ without guesswork.

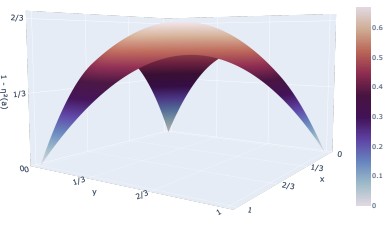 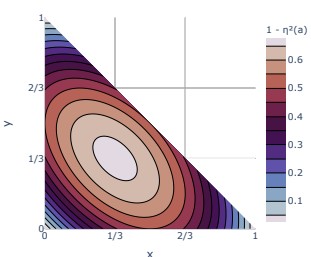

Figure 1: To visualize how $1 - \eta(S(\phi(\theta, x)))$ represents the confidence of a model's prediction we plot $1 - \eta(a)$ for all $a \in \mathbb{R}^3$ such that $a$ is a valid probability distribution over three classes (i.e. for all the elements of the standard 2-simplex). Since there are only two free variables, $x$ and $y$ in the plot represent $a$ by $a = [x, y, 1 - x - y]$. We see that $1 - \eta(a)$ is only zero when $a = e_i$ for some $i$, namely when a model would be most confident. We also note that $1 - \eta(a)$ is largest when a model would be least confident in its prediction–i.e. when $a = [1/3, 1/3, 1/3]$.

## 3.2 EXTENSION TO THE NON-LINEAR CASE

Now we will attempt to extend the derivation of the previous section to the non-linear case. This is a challenging undertaking so we will resort to numerical evidence. To extend the results from the affine case we will consider both the ACEHT and the upper bound $ACEHT(S(\phi(\theta, x)))) \leq |x|(1 - \gamma^2(S(\phi(\theta, x)), y))$ to which we refer as the "margin bound". We will compare both quantities to the empirically derived Hessian trace. To compute the empirical Hessian trace we use the PyHessian package (Yao et al., 2019) which implements Hutchinson's method (Bai et al., 1996; Avron & Toledo, 2011).

To compare the quantities we will compare them in terms of their distributions over the data. Specifically, let $(X, Y) \sim \mathcal{D}$ and fix $\theta$ then we compute the Pearson Correlation Coefficient (r-value) (Lee Rodgers & Nicewander, 1988) between the random variables $Tr(\mathbf{H}(\ell(\phi(\theta, X)), Y))$ and $ACEHT(S(\phi(\theta, X)))$ and similarly for the margin bound. The choice of the r-value is natural because in the affine case the ACEHT and the Hessian trace are equivalent, therefore a linear relationship should be expected. Our method is also more general than just comparing some statistic, such as the average (which is generally used for flatness measures), of the above random variables.

For example, while the smallest margin over the dataset is commonly used a generalization measure (Bartlett et al., 2017; Jiang et al., 2019; Neyshabur et al., 2017), Jiang et al. (2018) showed that higher moments of the distribution are a much better predictor for generalizability as we will also see in Section 4.

Figure 2 is an examples of such a fit for an affine predictor. While the high r-value of 0.97 confirms our analytic results, we also observe that the fit is not perfect, as would be expected due to the exact relationship. The inaccuracies are due to the numerical methods used and become more pronounced the higher the Hessian trace is. To avoid outliers heavily impacting the linear regression model in the non-linear case, we will use the SciPy function *LocalOutlierFactor* (Breunig et al., 2000) to remove outliers before fitting the line. With this we prevent hand picking points to skew the results and will also stabilize our results.

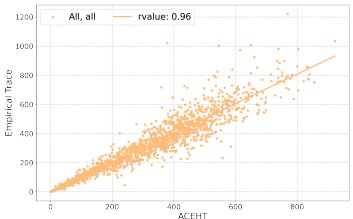

Figure 2: We consider the affine predictor $\phi((\theta, b), x) = \theta^t x + b$ with arbitrarily chosen parameters $(\theta, b)$ on $1,000$ randomly sampled datapoints from the MNIST dataset. Each scatter point represents a datapoint for which we compute the ACEHT and the empirical Hessian trace. The linear relationship between the ACEHT distribution and empirical Hessian trace both confirms our derivation of the ACEHT and provides a baseline for the numerical methods used for the empirical Hessian trace.

### 3.2.1 EMPIRICAL EVIDENCE

We present our results using the convolutional neural network LeNet on the MNIST dataset as they are representative of what we have observed on other architectures, hyperparameters, and datsets (see Appendix B). Our results use stochastic gradient descent with a fixed learning rate and batch size to achieve an appropriate performance on the classification task. Because of the computational difficulty of computing the empirical Hessian trace for every single element in the input data, we consider 1,000 randomly selected datapoints from the training-set. To highlight the computational difficulty of using even very optimized numerical tools, such as PyHessian, we note that it takes us roughly 1,5 hours to compute the Hessian trace for the whole MNIST dataset while it only takes 5 seconds for the margins.

In Figure 3 we present the plots related to the correlation of the empirical Hessian trace to the ACEHT and margin bound over the randomly sampled datapoints. Figures 3a and 3b show that for most of training, the correlation is between 0.8 and 1. Combining Figures 3a and 3c it can be seen that the r-value increases with the model training accuracy. Furthermore, the datapoint which are incorrectly predicted do not show a correlation.

With that we confirm the intuition that indeed, flatter solution are more robust and have larger margins. While we have found flatness and margins to be highly correlated in scenarios in which others have identified flatness to be a good generalization measure (Jiang et al., 2019; Keskar et al., 2016; Chaudhari et al., 2019), it may just be that this is also an epiphenomenon of stochastic gradient descent or some other process and there may be situations in which the relationship does not hold. However, our general advice to consider margins more is not impacted by this. In the scenario where generalizability and flatness have been linked, we have also shown that margins and flatness are correlated, hence it is advantageous to use margins instead due to computational reasons or for more complete intuition. The only situation in which it is more likely that margins and flatness are not correlated is when flatness has not yet been linked to generalizability. In such a situation it may also be better to use the better understood margin measure instead of using a flatness measure to assess generalizability. In the next section we will consider the first case, where we examine a general scenario in which flatness has been used to reason about generalizability and offer a more insightful margin perspective.

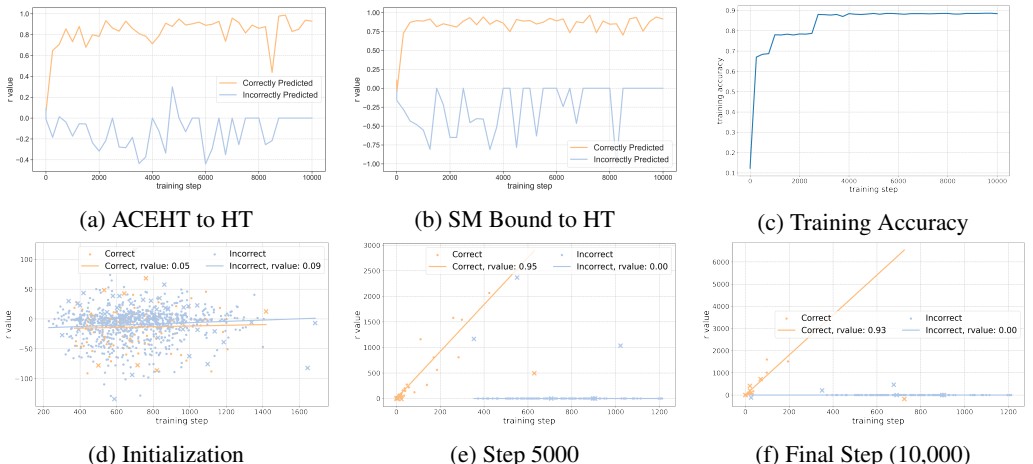

(a) ACEHT to HT          (b) SM Bound to HT          (c) Training Accuracy

(d) Initialization          (e) Step 5000          (f) Final Step (10,000)

Figure 3: We observe how ACEHT and the softmax margin (SM) bound relate, via the Pearson Correlation Coefficient, to the observed Hessian trace (HT) of LeNet trained on MNIST. In Figures 3a 3b we see that for correctly classified datapoints (orange) the empirical Hessian trace correlates well with the ACEHT and SM bound. In Figure 3c we observe that the increase in correlation occurs with an increase in training accuracy. To demonstrate the evolution of the distributions throughout training we plot the ACEHT and empirical HT distribution against each other in Figures 3d 3e 3f. We observe that while the most apparent outliers were removed, some still skew the linear regression.

## 4 PERSPECTIVE ON LARGE AND SMALL-BATCH METHODS

We now show how our results lead to a better understanding of phenomena which have been misleadingly attributed to flat minima. To do so, we consider the experiments which rekindled the debate around flat minima by Keskar et al. (2016), where flatness was used to explain why small-batch methods tend to generalize better than large-batch methods. The idea was that small-batch methods converge to flatter minima due to them being able to "escape" sharp minima more easily. However, it has been shown that the minima of both methods appear to be in the same attractive basin (Sagun et al., 2017; Freeman & Bruna, 2016; Draxler et al., 2018), meaning that small-batch methods do not seem to escape any attractive basin but are merely in a different area of the same attractive basin. While the results gave credence to flatter minima generalizing better, flatter minima do not seem to provide the full picture for why large-batch methods tend to do worse and we believe that an explanation in terms of the margins is more illuminating.

### 4.1 EXPERIMENT SETUP

We will replicate the experiment by Keskar et al. (2016) for a fully connected network with batch-normalized layers on the MNIST dataset as described in Appendix A. We chose the large-batch size to be 4096 and the small-batch size to be 256. To have a fair comparison, we use the same seed and take 10,000 gradient steps for both methods, instead of basing the stopping time on epochs. We also used stochastic gradient descent without Momentum. With our setup we observe a similar phenomenon as Keskar et al. (2016) in Table 1. The small and large-batch method both attain the same training accuracy and comparable training loss. However, the small-batch method is at a considerably flatter minimum and generalizes better than the large-batch method. We will now show that instead of considering the flatness, it would be more insightful to consider margins to explain the difference in generalizability.

While the upper bound of ACEHT is in terms of the softmax margins, we consider the output margins in this section. The reason is that most margin based generalization measures use the output margins. Another more practical reason is that towards the end of training, the softmax margins are all very close to 1 making it difficult to visualize and observe the distribution. We also do not use a normalized version of the margins (such as Bartlett et al. (2017); Jiang et al. (2018)). Our reasoning

is that because we use the same architecture, the same dataset, and train in a similar manner the margin distributions will be comparable.

|  | Loss | | Accuracy | | Trace |
| --- | --- | --- | --- | --- | --- |
| Batch Size | Training | Test | Training | Test | Training |
| 256 | $2.1 \times 10^{-5}$ | $7.28 \times 10^{-2}$ | 1.0 | 0.9834 | 1.01 |
| 4096 | $3.4 \times 10^{-5}$ | $9.45 \times 10^{-2}$ | 1.0 | 0.9794 | 6.62 |

Table 1: The results of our trained fully connected network with batch-normalized layers on MNIST optimized with SGD and a 0.1 learning rate. The results reflect the observations made by Keskar et al. (2016). I.e. the small-batch method has a smaller Hessian trace and generalizes better.

## 4.2 A Margin Perspective on Large and Small-Batch Sizes

In Figure 4 we see that the output margins and the Hessian trace are correlate as expected from Section 3. We can also roughly see that the small-batch method has fewer low margins than the large-batch methods. To emphasize this difference we consider Figure 4c where we plot the histogram and box-plot of the output margin distribution for both the large-batch and small-batch method. We also display the skewness of each, which is the third moment centered around the mean. The box-plots and the skewness confirm that the small-batch method is dominated by large margins indicating better generalizability (as discussed in Bartlett et al. (2017); Jiang et al. (2018)). The idea with a left-skewed margin distribution is that the tail with low margin datapoints is mostly compromised of outliers and will not massively affect the robustness to input perturbations. This soft-margin SVM perspective is in contrast to hard-margin SVMs where the margin is defined to be the minimum of all the distances to the decision boundary (Shalev-Shwartz & Ben-David, 2014). If a hard-margin view was adopted, then the small-batch method would be predicted to generalize worse, because it has the smallest margin as we see in Figure 4c. However, the distribution of the small-batch method is also more left skewed, which would point to this minimum being an outlier rather than being indicative of generalizability.

We now want to explain why the small-batch method generalizes well. As observed in Jastrzębski et al. (2017) a smaller batch-size is similar to a larger learning-rate, hence at every step the process will advance further than a large batch-method would. It has already been noted by Hoffer et al. (2017) that training longer leads the large-batch method to generalize just as well as the small-batch method because it had time to "catch up", even though the decrease in training loss may be barely noticeable. We have also seen that SGD converges to margin maximizing solutions by Banburski et al. (2019). Therefore, a method that is able to train or advance further, will also be closer to a margin maximizing solution. We therefore expect that large-batch methods not having had enough time to maximize margins is the driving force behind the large vs small-batch phenomenon.

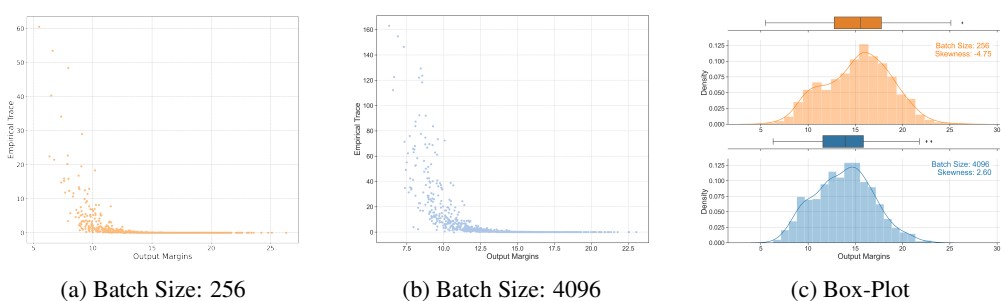

| (a) Batch Size: 256 | (b) Batch Size: 4096 | (c) Box-Plot |
| --- | --- | --- |

Figure 4: We plot both small-batch method (orange) and large-batch method (blue). In Figures 4a and 4b we plot the output margins against the Hessian trace for each datapoint. We observe a strong relationship between the Hessian trace and the output margins. In Figure 4c we plot both the histogram and box-plot and display the skewness (the third standardized moment) for both the large and small-batch method's margin distributions. We observe that the distribution of the small-batch method is more left skewed which would indicate better generalizability independent of the flatness.

## 5  BECOMING FLATTER WITH INCREASING MARGINS

Reparametrization problems such as shown by Dinh et al. (2017) are neither a new phenomenon nor should they necessarily discourage the design of algorithms which attempt to find flat minima. Rather they inform on what aspects of a generalization measure need to be adjusted to allow them to be used in a practical setting. For SVMs, the problem of scaling the hyperplane normal to increase margins of correctly classified points is solved by scaling the normal to make it a unit vector, transforming the functional margin into the geometric margin (Shalev-Shwartz & Ben-David, 2014). In the case of neural networks, it is also known that scaling the last layer leads to an increase in the margins for data which has been correctly predicted (Neyshabur et al., 2017). This scaling issues has been successfully addressed (see Bartlett et al. (2017); Elsayed et al. (2018); Jiang et al. (2018)). Due to the relationship to the classification margins it is natural to ask if flatness suffers from a similar problem. We confirm this with the following Proposition:

**Proposition 5.1.** *For a given neural network $\phi$ let $T_\alpha : \Theta \to \Theta$ be such that for all $x \in \mathcal{X}$ and $\theta \in \Theta$ we have $\phi(T_\alpha(\theta), x) = \alpha\phi(\theta, x)$. Now assume that $\theta' \in \Theta$ and a datapoint $(x', y')$ for which $argmax_{k \in \{1,...,C\}}(\phi(\theta, x'))_k = y'$ then*

$$\forall s, t \in \{1, ..., dim(\Theta)\} \lim_{\alpha \to \infty} \partial_{\theta_s} \partial_{\theta_t} \ell(\phi(T_\alpha(\theta'), x'), y') = 0. \tag{1}$$

The proof is in the Appendix D. From the Proposition we immediately derive the following Corollary:

**Corollary 5.2.** *Assume that $\phi$ and $\theta$ predict every datapoint in a set $D$ correctly then*

$$\forall s, t \in \{1, ..., dim(\Theta)\} \lim_{\alpha \to \infty} \partial_{\theta_s} \partial_{\theta_t} \mathcal{L}_D(T_\alpha(\theta)) = 0. \tag{2}$$

Due to the Corollary, if a network has achieved full training accuracy, then the network is equivalent under the $T_\alpha$ transformation to an arbitrarily flat network. We note that there exists such a $T_\alpha$ transform for most networks. Scaling the last layer is one simple instance of such a transform. Another is that for fully connected and convolutional networks with ReLU non-linearities we observe that by the non-negative homogeneity scaling each layer also results in a valid $T_\alpha$ transformation. The crucial property of the $T_\alpha$ map is that it does not change the relative order of the model outputs and therefore, given two networks which have achieved full training accuracy we can not determine which network should generalize better based solely on the flatness of the local-geometry. We note that Banburski et al. (2019) mentioned such an issue but they did not discuss it in the context of flat minima and their arguments relied on further structure which we believe is less illuminating than our presentation and proofs.

## 6  CONCLUSIONS

In this paper, we have related flatness to the classification margins in a principled manner, in contrast to other works that have made a more intuitive or less quantifiable connection (Huang et al., 2019; Wang et al., 2018; Neyshabur et al., 2017; Petzka et al., 2020). Our results lead to the immediate practical recommendation of using margins instead of the computationally expensive flatness to assess generalizability. We also use our results to replace the misleading notion that small-batch methods generalize better because they "escape" sharp minima, instead arguing that small-batch methods have more time to maximize margins. We were also motivated by the flatness and margin relationship to highlight that neural networks can be made arbitrarily flat. This implies that the generalizability of two networks can not be distinguished based on flatness and hence needs to be addressed to make flatness a viable generalization measure. Based on our results, future work may assess whether flatness is an epiphenomenon of the optimization methods, because now recent work on margins (e.g. Banburski et al. (2019); Soudry et al. (2018)) can be applied to reason about flatness. Furthermore, by relating properties of the parameter space (flatness) to properties of the input space (margin) there is now an opportunity to further explore results such as by Sagun et al. (2017), where they found that the Hessian, with respect to the parameters of a neural network upon convergence, has as many positive eigenvalues as the number of classes in the dataset used. Overall, our results enable more principled discussion on how flatness may contribute to generalizability.

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

## A    APPENDIX: NETWORK ARCHITECTURE AND DATASETS

### A.1    NETWORK ARCHITECTURE

We implement the convolutional neural network LeNet-5 as described in LeCun et al. (1998). Our fully connected neural network with batch normalized layers (FCNBN) is inspired by Keskar et al. (2016). It has a 784-dimensional (MNIST) or 1024-dimensional (CIFAR10) input layer followed by three batch-normalized Ioffe & Szegedy (2015) layers with ReLU non-linearities and a 10-dimensional output layer.

### A.2    DATASETS

| Dataset | # Training Points | # Test Points | Features | # Classes |
|---|---|---|---|---|
| MNIST (LeCun et al., 1998) | 60,000 | 10,000 | $28 \times 28$ | 10 |
| CIFAR10 (Krizhevsky et al., 2009) | 50,000 | 10,000 | $32 \times 32$ | 10 |

Table 2: The datasets used for this paper.

## B    APPENDIX: FLATNESS AND MARGIN CORRELATION

Here we present further evidence of the flatness and margin correlation discussed in Section 3. Like in Section 3 we have used appropriate learning rates and batch sizes to get a reasonable performance for the task, and have observed our results to hold for different hyperparameters. One instance where we demonstrate two different batch-sizes is for the Fully Connected Network with Batch Normalization on MNIST (Section B.1) where we present results for a batch size of 256 and 4096. We again only consider 1,000 randomly selected datapoints from the training-set due to the computational difficult of computing the Hessian trace. If the network achieves full training accuracy and there are no incorrectly classified datapoints, we set the r-value to zero.

Overall, we observe the same results as in Section 3 and a correlation between 0.8 and 1. As before, the correlation increases with increasing training accuracy for correctly predicted datapoints.

## B.1 FULLY CONNECTED NETWORK WITH BATCH NORMALIZATION ON MNIST

### B.1.1 BATCH SIZE: 256

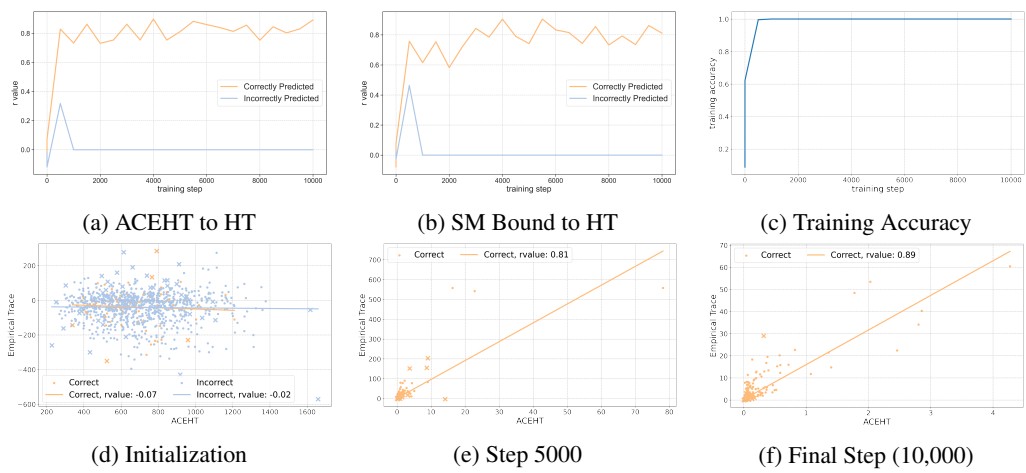

(a) ACEHT to HT     (b) SM Bound to HT     (c) Training Accuracy

(d) Initialization     (e) Step 5000     (f) Final Step (10,000)

### B.1.2 BATCH SIZE: 4096

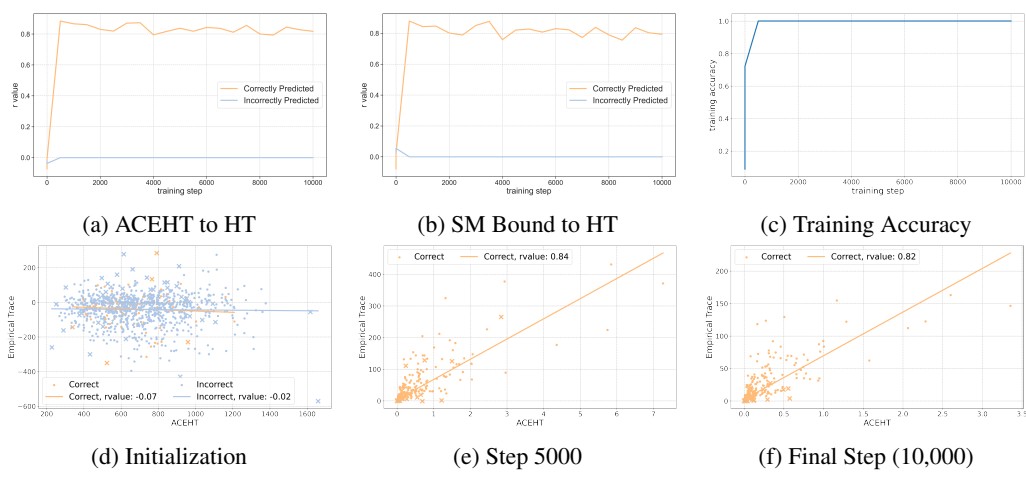

(a) ACEHT to HT     (b) SM Bound to HT     (c) Training Accuracy

(d) Initialization     (e) Step 5000     (f) Final Step (10,000)

## B.2 LENET ON CIFAR10

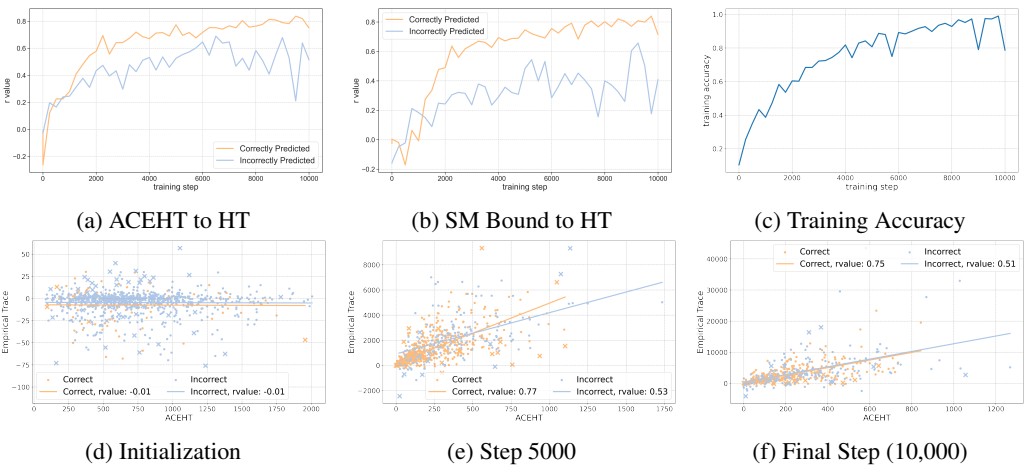

(a) ACEHT to HT     (b) SM Bound to HT     (c) Training Accuracy

(d) Initialization     (e) Step 5000     (f) Final Step (10,000)

### B.2.1 Fully Connected Network with Batch Normalization on CIFAR10

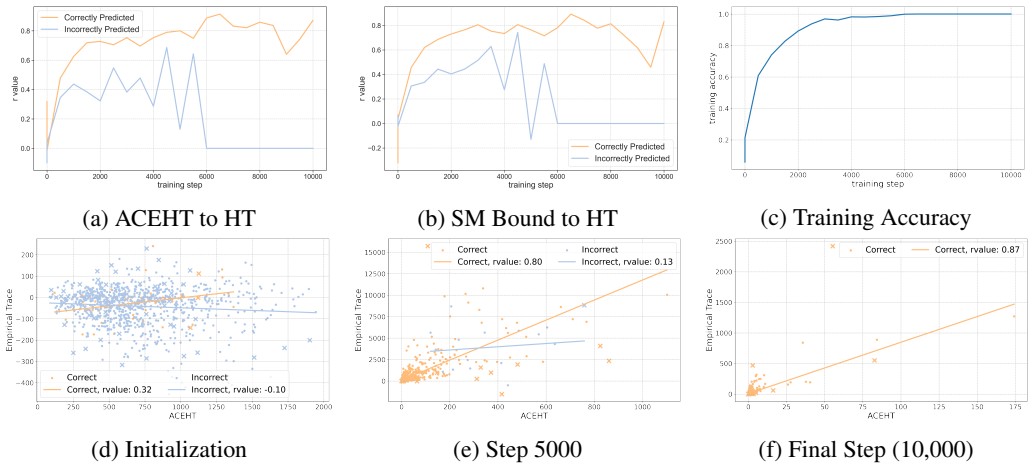

(a) ACEHT to HT  (b) SM Bound to HT  (c) Training Accuracy

(d) Initialization  (e) Step 5000  (f) Final Step (10,000)

## C  Appendix: Derivatives of the Cross-Entropy Loss

### C.1  General Form

For the general form we consider the cross-entropy loss for a predictor function which is scaled by some scalar $\alpha$. Specifically, we assume an arbitrary input-output pair $(x, y) \in \mathcal{X} \times \mathcal{Y}$ and will compute the partial derivatives with respect to the parameters $\theta$ of the predictor function $\alpha\phi(\theta, x)$. Since the equations can become very long we will declutter the notation by letting $S = S(\alpha\phi(\theta, x))$, $\phi = \phi(\theta, x)$ and for two d-dimensional vectors $x, y \in \mathbb{R}^d$ we write $\langle x, y \rangle = \sum_{i=1}^{d} x_i y_i$. We also denote elementwise multiplication by $\odot$ and let $\Phi$ be a matrix such that $(\Phi)_{ij} = \phi_j$.

**Lemma C.1.** *The first partial derivative of the cross-entropy loss with respect to an element $\theta_i$ is:*

$$\partial_{\theta_i} \ell(\alpha\phi(\theta, x), y) = -\alpha(\partial_{\theta_i}\phi_y - \sum_{l=1}^{C} \partial_{\theta_i}\phi_l S_l(\phi))$$
$$= -\alpha(\partial_{\theta_i}\phi_y - \langle \partial_{\theta_i}\phi, S \rangle).$$

*Proof.*

$$\partial_{\theta_i} \ell(\alpha\phi, y) = \partial_{\theta_i} - log(S_y)$$
$$= -\frac{1}{S_y}\partial_{\theta_i} S_y \tag{3}$$

With some manipulation we compute $\partial_{\theta_i} S_y$:

$$\partial_{\theta_i} S_y = \frac{\partial_{\theta_i} e^{\alpha\phi_y}}{\sum_{k=1}^{C} e^{\alpha\phi_k}} - \frac{e^{\alpha\phi_y}}{(\sum_{k=1}^{C} e^{\alpha\phi_k})^2} \sum_{l=1}^{C} \partial_{\theta_i} e^{\alpha\phi_l}$$
$$= \alpha\partial_{\theta_i}\phi_y \frac{e^{\alpha\phi_y}}{\sum_{k=1}^{C} e^{\alpha\phi_k}} - \alpha\frac{e^{\alpha\phi_y}}{\sum_{k=1}^{C} e^{\alpha\phi_k}} \sum_{l=1}^{C} \frac{e^{\alpha\phi_l}}{\sum_{k=1}^{C} e^{\alpha\phi_k}} \partial_{\theta_i}\phi_l$$
$$= S_y\alpha(\partial_{\theta_i}\phi_y - \sum_{l=1}^{C} \partial_{\theta_i}\phi_l S_l). \tag{4}$$

Combining Equations 3 and 4 we obtain Lemma C.1:

$$\partial_{\theta_i} \ell(\alpha\phi, y) = -\alpha(\partial_{\theta_i}\phi_y - \sum_{l=1}^{C} \partial_{\theta_i}\phi_l S_l).$$

$\square$

**Lemma C.2.** *The second partial derivative of the cross-entropy loss with respect to elements $\theta_s$ and $\theta_t$ is:*

$$\partial_{\theta_t}\partial_{\theta_s}\ell(\alpha\phi(\theta,x),y) = -\alpha(\partial_{\theta_t}\partial_{\theta_s}\phi_y - \langle\partial_{\theta_t}\partial_{\theta_s}\phi, S\rangle - \langle\partial_{\theta_s}\phi, \alpha S \odot (\partial_{\theta_t}\phi - (\partial_{\theta_t}\Phi)S)\rangle).$$

*Proof.* Differentiating the first order derivative given by Lemma C.1 we obtain by the multi-variable chain rule:

$$\partial_{\theta_t}\partial_{\theta_s}\ell(\alpha\phi,y) = -\alpha(\partial_{\theta_t}\partial_{\theta_s}\phi_y - \langle\partial_{\theta_t}\partial_{\theta_s}\phi, S\rangle - \langle\partial_{\theta_s}\phi, \partial_{\theta_t}S\rangle). \tag{5}$$

To compute $\partial_{\theta_t}S$ in Equation 5 we use Equation 4 and obtain:

$$
\begin{aligned}
(\partial_{\theta_t}S)_i &= \partial_{\theta_t}S_i \\
&= S_i\alpha(\partial_{\theta_t}\phi_i - \langle\partial_{\theta_t}\phi, S\rangle),
\end{aligned}
$$

which after some simplification reduces to:

$$\partial_{\theta_t}S(\phi) = \alpha S \odot (\partial_{\theta_t}\phi - (\partial_{\theta_t}\Phi)S). \tag{6}$$

Combining Equations 5 and 6 we obtain Lemma C.2:

$$\partial_{\theta_t}\partial_{\theta_s}\ell(\alpha\phi,y) = -\alpha(\partial_{\theta_t}\partial_{\theta_s}\phi_y - \langle\partial_{\theta_t}\partial_{\theta_s}\phi, S\rangle - \langle\partial_{\theta_s}\phi, \alpha S \odot (\partial_{\theta_t}\phi - (\partial_{\theta_t}\Phi)S)\rangle).$$

$\square$

## C.2 AFFINE CROSS-ENTROPY HESSIAN TRACE

We now present the proof of Proposition 3.1:

*Proof.* Throughout the proof we make use of Lemma C.2 and let $\alpha = 1$. We also notice that any second derivative with respect to $\phi((\theta,b),x)$ is zero since $\phi$ is an affine classifier.

We first consider the derivatives with respect to elements of $\theta$ where we use $\theta_{i,j}$ to denote the element in the $i$th row and $j$th column of the matrix $\theta$. Notice that $\partial_{\theta_{i,j}}\phi = x_j e_i$ which we write as $x_j^i$. The second order derivatives are given by:

$$
\begin{aligned}
\partial_{\theta_{i,j}}\partial_{\theta_{s,t}}\ell(\phi,y) &= \langle x_t^s, S \odot (x_j^i - x_j S_i \mathbf{1})\rangle \\
&= -x_t(S_s((x_j^i)_s - x_j S_i)),
\end{aligned}
$$

when computing the trace we only compute the elements on the diagonal and hence we get:

$$
\begin{aligned}
\partial_{\theta_{i,j}}\partial_{\theta_{i,j}}\ell(\phi,y) &= x_j(S_i(x_j - x_j S_i)) \\
&= x_j^2 S_i(1 - S_i)
\end{aligned}
$$

Now we consider derivatives with respect to elements of $b$ and notice that $\partial_{b_i}\phi = e_i$. For the second derivative we then get:

$$
\begin{aligned}
\partial_{b_i}\partial_{b_j}\ell(\phi,y) &= \langle e_j, S \odot (e_i - e_i S_i)\rangle \\
&= \delta_{ij}S_i(1 - S_i).
\end{aligned}
$$

Finally, summing up the diagonal of the total Hessian we get:

$$Tr(H(l(\Theta,x,y))) = (|x|^2 + 1)(1 - \sum_j S_j^2)$$

where we used the fact that $\sum_i S_i = 1$.

$\square$

# D  APPENDIX: SCALING PROOF

To prove Proposition 5.1 we first prove the following lemma:

**Lemma D.1.** *Assume that the argmax of $\phi$ is the correct class $y$ and is unique then for $k \in \mathbb{N}, k \geq 1$ and $i \neq y$:*

$$\lim_{\alpha \to \infty} \alpha^k S_i(\alpha\phi) = 0 \tag{7}$$

*Proof.* Let $y$ be such that $\phi_y = max_{k \in \{1,...,C\}}\phi_k$. For $i \neq y$ we have:

$$\lim_{\alpha \to \infty} \alpha^k \frac{e^{\alpha\phi_i}}{\sum_{k=1}^{C} e^{\alpha\phi_k}} = \lim_{\alpha \to \infty} \frac{\alpha^k}{e^{\alpha(\phi_y - \phi_i)} + \sum_{k=1,k\neq y}^{C} e^{\alpha(\phi_k - \phi_i)}}$$

$$= \lim_{\alpha \to \infty} \frac{k!}{(\phi_y - \phi_i)^k e^{\alpha(\phi_y - \phi_i)} + \sum_{k=1,k\neq y}^{C}(\phi_k - \phi_i)^k e^{\alpha(\phi_k - \phi_i)}}$$

where the last line follows from applying L'Hopital's rule k times. Since we assumed that $y$ is the only $y \in \{1, ..., N\}$ such that $\phi_y = \max_{k \in \{1,...,N\}} \phi_k$, we have that $\phi_k < \phi_y$ for all $k \neq y$. Hence, as $\alpha \to \infty$ we have $e^{\alpha(\phi_k - \phi_y)} \to 0$. Therefore:

$$\lim_{\alpha \to \infty} \frac{k! e^{\alpha(\phi_i - \phi_y)}}{(\phi_y - \phi_i)^k + \sum_{k=1,k\neq y}^{C}(\phi_k - \phi_i)^k e^{\alpha(\phi_k - \phi_y)}} = 0$$

$\square$

We are now ready to prove Proposition 5.1:

*Proof.* We first show that the term $-\alpha^2 \langle \partial_{\theta_i}\phi, S \odot (\partial_{\theta_t}\phi - \langle \partial_{\theta_t}\Phi, S \rangle) \rangle$ always goes to zero. Expanding we get:

$$\alpha^2 \left( \sum_{l=1}^{C} \partial_{\theta_i}\phi_l (S_l(\partial_{\theta_t}\phi_l - \sum_{k=1}^{C} \partial_{\theta_t}\phi_k S_k)) \right)$$

$$= \alpha^2 \partial_{\theta_i}\phi_y S_y(\partial_{\theta_t}\phi_l - \sum_{k=1}^{C} \partial_{\theta_t}\phi_k S_k) + \alpha^2 \left( \sum_{l=1,l\neq y}^{C} \partial_{\theta_i}\phi_l (S_l(\partial_{\theta_t}\phi_l - \sum_{k=1}^{C} \partial_{\theta_t}\phi_k S_k)) \right)$$

We now show that each term in the sum goes to zero. Consider $l \neq y$:

$$|\alpha^2 \sum_{k=1}^{C} \partial_{\theta_i}\phi_l S_l(\partial_{\theta_t}\phi_l - \partial_{\theta_t}\phi_k S_k)| \leq \alpha^2 S_l \sum_{k=1}^{C} |\partial_{\theta_i}\phi_l(\partial_{\theta_t}\phi_l - \partial_{\theta_t}\phi_k S_k)| \text{by the Triangle Inequality and } 0 \leq S_l \leq 1$$

$$\leq \alpha^2 S_l C$$

We let $M = \sum_{k=1}^{C} |\partial_{\theta_i}\phi_l(\partial_{\theta_t}\phi_l - \partial_{\theta_t}\phi_k S_k)|$ and note that $M < \infty$ for all $0 < \alpha < \infty$ since $0 \leq S_k \leq 1, \partial_{\theta_i}\phi_l, \partial_{\theta_t}\phi_l, \partial_{\theta_t}\phi_k$ are constants, and it is a finite sum. By Lemma D.1 as $\alpha \to \infty$ we have $\alpha^2 S_l C \to 0$ and hence $\alpha^2 \sum_{k=1}^{C} \partial_{\theta_i}\phi_l S_l(\partial_{\theta_t}\phi_l - \partial_{\theta_t}\phi_k S_k) \to 0$.

We now consider $l = y$:

$$|\alpha^2 \partial_{\theta_i}\phi_y S_y(\partial_{\theta_t}\phi_y - \sum_{k=1}^{C} \partial_{\theta_t}\phi_k S_k)| \leq \alpha^2 |\partial_{\theta_i}\phi_y| S_y \left( |\partial_{\theta_t}\phi_y - \partial_{\theta_t}\phi_y S_y)| + \sum_{k=1,k\neq y}^{C} |\partial_{\theta_t}\phi_k S_k| \right)$$

$$\alpha^2 |\partial_{\theta_t}\phi_y - \partial_{\theta_t}\phi_y S_y)| = |\partial_{\theta_t}|\alpha^2 T|\frac{\sum_{s=1,s\neq y}^{C} e^{\alpha\phi_s}}{\sum_{m=1}^{C} e^{\alpha\phi_m}}|$$

$$= |\partial_{\theta_t}|\alpha^2 \sum_{s=1,s\neq y}^{C} S_s \text{ since } S_s > 0$$

and using similar arguments and Lemma D.1 follows that this term is zero in the limit.

It is also obvious that $\alpha^2 \sum_{k=1,k\neq y}^{C} |\partial_{\theta_t}\phi_k S_k|$ goes to zero.

We are left with showing that $\alpha(\partial_{\theta_t}\partial_{\theta_i}\phi_y - \langle\partial_{\theta_t}\partial_{\theta_i}\phi, S\rangle)$ goes to zero, this is only guaranteed when $y$ is the true label). We will use the same method as above:

$$|\alpha(\partial_{\theta_t}\partial_{\theta_i}\phi_y - \langle\partial_{\theta_t}\partial_{\theta_i}\phi, S\rangle)| \leq \alpha(|\partial_{\theta_t}\partial_{\theta_i}\phi_y - \partial_{\theta_t}\partial_{\theta_i}\phi_y S_y| + \sum_{l=1,l\neq y}^{C} |\partial_{\theta_t}\partial_{\theta_i}\phi_l S_l|)$$

and the result follows using again Lemma D.1. $\qquad\square$

