# OpenReview forum: "On Flat Minima, Large Margins and Generalizability"
_ICLR.cc/2021/Conference — Reject_

### Official Review · AnonReviewer3 · 2020-10-26
**Empirical Relation between the Softmax Output Margin and Flatness of the Loss Surface**

**Rating:** 4
**Confidence:** 4

**Review:**

The paper presents empirical evidence that the output margin - as a measure of the confidence of a multiclass predictor - is strongly correlated to the Hessian trace when using cross-entropy loss with softmax. Moreover, the paper presents a method for estimating the Hessian trace using the input norm and softmax output. This estimation is inspired by linear classifiers and shows a strong correlation with the Hessian trace.

I think the paper presents an interesting take on generalization by relating flatness and output margins. I would argue that it might be even more fruitful to also investigate input margins, or margins in some representation (i.e., some layer of the network). Nonetheless, the direction is interesting and the paper is well-written and pleasant to read. However, the analysis is a bit limited and the claimed contributions of this paper are not well substantiated.

The paper claims that it proves that flatness and output margins are strongly correlated. The paper proves this relation for linear models with cross-entropy loss and softmax, but only presents empirical indication for neural networks. While those experiments are interesting, they are very limited (only MNIST). Thus, they do not fully substantiate the claim.

Moreover, the paper claims to show that small-batch SGD is not better in escaping local minima than large-batch SGD. To that end, the paper performs experiments in which small-batch SGD indeed generalizes better than large-batch SGD. The paper explains this by the distribution of output margins. This alone does not support the claim. The paper than argues that Hoffer et al., 2017 have shown that large-batch training generalizes just as well (given a similar number of parameter updates) and from that concludes their claim is true. There are two issues with this argument: (i) it remains unclear whether the empirical results of Hoffer et al., 2017 show that large-batch training always generalizes as well as small-batch training given the same number of weight updates, since Hoffer already found that the generalization is also connected to their batch normalization technique and other works (e.g., Goyal et al, 2017 [1]) report that large-batch training generalizes not as well as small-batch training. (ii) even if Hoffer et al., 2017 would support the claim that large-batch training generalizes as well as small-batch training, I don't see why the arguments made by the authors add to this. To give compelling empirical evidence the paper could, for example, show that the distribution of margins is different for small and large batch training over a wide selection of different datasets and network architectures. This would not prove the claim, but make it more plausible. In general, I am unsure whether this claim can be true for arbitrary learning problems. The argument made for small-batch training is that SGD in expectation optimizes the risk, while gradient descent optimizes the empirical error (see Chapter 14.5.1 in Shalev-Shwartz and Ben-David, 2014). Thus, using a small-batch size (and thus performing something closer to SGD) should optimize the true risk and thus generalize better than using a large-batch (thus being closer to GD) which should overfit. Of course, it is unclear how this observation on SGD translates to neural networks which often are trained in the interpolation regime. However, this paper in its current form does not add insights to this discussion.

Lastly, the paper claims to show that neural networks can be scaled to be arbitrary flat and that this issue relates to the output margin. The scaling issue itself is well-known (Dinh et al., 2917) and reparameterization-invariant flatness measures have been proposed (Petzka, et al., 2019, 2020, Tsuzuku, et al. 2019) that avoid this issue. The difference of those measures to the one proposed in this paper is not discussed. Thus, the contribution of this paper remain unclear.

Therefore, I think this paper is not ready for publication, yet.

Detailed comments:

- it seems that the major advantage of using the margin measure instead of the Hessian trace is computational complexity. If this is indeed the selling point, please add an analysis of the computational complexity and a runtime analysis (e.g., when regularizing using the measure). In that case, please also compare to flatness measures of only one layer of the network (e.g., Petzka et al., 2020).
- it seems that the margin measure does not suffer from the reparameterization curse which is a huge plus for this method. I suggest adding a short section that proves this.
- in Sec. 4.2 the paper relates its results on output margins to input margins (mentioning SVMs). It is unclear to me how output margins relate to input margins. I do think that this relation is interesting, but the brief discussion in that section is not compelling.

[1] Goyal, Priya, et al. "Accurate, large minibatch sgd: Training imagenet in 1 hour." arXiv preprint arXiv:1706.02677 (2017).

------- After Discussion --------
The authors agreed that the paper is not ready for publication, yet. Thus I keep my original score.

---

> ### Comment · AnonReviewer3 · 2020-11-24
> **No rebuttal**
>
> Since the authors have not written a rebuttal, I will keep my original score.

---

### Official Review · AnonReviewer2 · 2020-10-28
**An interesting collection of insights connecting flatness and margins; but concerned about novelty**

**Rating:** 4
**Confidence:** 3

**Review:**

**Update**: Since there's no substantial author response, I'm keeping my score as it is. All the best for your future submissions!

### Summary of paper

- This paper argues that the flatness of a deep network at an input --- quantified by the Hessian trace of the cross-entropy loss --- roughly corresponds to the margin of the network on that point. The paper argues this by first explicitly deriving a relation between the two for a linear classifier. Then, for non-linear networks, it demonstrates this relation empirically via correlations between the two terms, on two architectures (LeNet and fully connected network) and datasets (MNIST and CIFAR10).

- Then the paper uses the above insight to explain why large batch methods appear to generalize worse than small batch methods. The observation is that the margin of small batch methods are on average larger than small batch methods, and this implies that the large batch method has not trained long enough to maximize its margins and generalize well.

- Finally, the paper shows that rescaling the weights of the network can lead to arbitrarily flat loss landscapes and arbitrarily large minima.

### Strengths
The paper attempts to make a variety of interesting points connecting flatness and margins. The claims made are easy to understand, and are supported by a variety of illuminating experiments.

### Weaknesses
0. The paper violates the ICLR margins. Would be great if the authors can resubmit a version of the paper with the correct margins.

1. The argument that "flatness is related to margins" as such I'm afraid is not novel to me and has appeared in PAC-Bayesian analyses. Here, one thinks of flatness in terms of how much random perturbation a trained deep network can withstand until a non-negligible fraction of the training data end up being misclassified. If the training data is classified by a larger margin, the classifier would withstand larger perturbations, and hence implying more flatness. For example Neyshabur et al., '18 has a nice formalization of this. Having said that, I agree that the relationship between the trace of the Hessian of cross-entropy loss to its margin is strictly speaking, novel. However, I'm not sure I see much new insight here since this insight is claimed to be a fundamental contribution of this paper.


2. There are many existing "normalized" flatness measures, which I feel this paper seems should have engaged with more deeply than it has. In particular, this paper argues that the trace of the Hessian or the margin of the deep network can be scaled arbitrarily without affecting the performance of the network -- therefore these quantities in themselves are bad measures of generalization. Isn't this exactly the point made in Neyshabur et al., '17,18, Bartlett et al., '17 etc., in the context of generalization bounds, and in Dinh et al., '17 in the context of Keskar et al., '17's work? The argument as to what is fundamentally new in this paper when compared to those, is lacking.

3. I'd also want to note that Hessian-based flatness measures capture only limited local information given that the landscape is non-convex and we've ReLU units. Would be nice to note this in the paper. (This is however not a problem with PAC-Bayesian measures).

4. Unfortunately, the section explaining why small-batch training generalizes better was not any more illuminating to me than what we understood from Hoffer et al 2017 and Jastrze ̨bski et al. (2017).

5. I'd also like to note that  "Flatness is a False Friend", Diego Granziol
https://arxiv.org/abs/2006.09091 makes a similar point about how training longer naturally leads to flatter minima in terms of the Hessian. Depending on how concurrent the authors think that work is, the authors may want to consider citing it.

### Overall opinion
While understanding connections between flatness and generalization is an important direction, unfortunately, I feel that this paper does not provide significantly new insights into these terms. Hence, I'd recommend rejection at this point.


### Clarification questions

I'm not sure I understood the architecture that is used in Fig 2. Have you trained a linear classifier?

#### Minor suggestions

- Page 4 "Fig 2 is an examples of"

#### References

- A PAC-Bayesian Approach to Spectrally-Normalized Margin Bounds for Neural Networks ICLR 2018
Behnam Neyshabur, Srinadh Bhojanapalli, Nathan Srebro https://arxiv.org/pdf/1706.08947.pdf

- Nitish Shirish Keskar, Dheevatsa Mudigere, Jorge Nocedal, Mikhail Smelyanskiy, and Ping Tak Peter Tang. On large-batch training for deep learning: Generalization gap and sharp minima. arXiv preprint arXiv:1609.04836, 2016.

- Yiding Jiang, Behnam Neyshabur, Hossein Mobahi, Dilip Krishnan, and Samy Bengio. Fantastic generalization measures and where to find them. arXiv preprint arXiv:1912.02178, 2019.

- Laurent Dinh, Razvan Pascanu, Samy Bengio, and Yoshua Bengio. Sharp minima can generalize for deep nets. In Proceedings of the 34th International Conference on Machine Learning-Volume 70, pp. 1019–1028. JMLR. org, 2017.

---

### Official Review · AnonReviewer4 · 2020-10-28

**Rating:** 4
**Confidence:** 4

**Review:**

----------- Overview of the paper --------------

This paper studies the correlation of flat minima and large margins, as well as their impact on the generalization abilities of neural networks. The main tool in the main is the relation between the trace of Hessian and the functional margin. The empirical study in the paper reveals that both in linear and nonlinear settings, there exists an intimate connection between the trace of Hessian and the margin. This idea is further utilized to explain learning with large/small batch sizes.

----------- Contributions and strength--------------

Properties of the stationary point found by training algorithms (e.g., SGD) are sweet spot topics, especially in deep learning. Flatness and the corresponding margins are believed to be influential factors on the generalizability of neural networks. The paper makes efforts in explaining the interplay of these two factors, which is helpful in forming a deep understanding of neural networks. The message in the paper is rather clear.

----------- Weakness --------------

Some claims in the paper are conjectures and lack a serious treatment. For example, in section 4, the margin perspective on large and small-batch sizes are based on some folklore and beliefs without empirical/theoretical confirmation. The experiments in section 4 is used to support the observation in existing works, rather not for further explanation of the observation. Extra experiments corroborate that ``large-batch methods not having had enough time to maximize margins is the driving force behind the large-batch vs small-batch phenomenon'' should be provided.

For the theory part, there seem to be some typos.

----------- Questions and comments --------------

In proposition 3.1 and the following discussion, \eta is undefined or it should be replaced by \gamma. It may be better to write the bound using the margin in the proposition statement. I am also curious why there is a dependence on the norm of input in the trace of the Hessian, and what are the implications of such a dependence?

I am a bit confused of the message in section 5. The concluding remark seems to argue that when the data is perfectly classified, simple scaling can alter the margin yet does not change the generalization. I think this is a direct consequence of the positive homogeneous property of classifiers.

The paper is prepared in a nonstandard ICLR template. The margin of the current draft is much smaller than the ICLR template. Although the content is fit in 8 pages, it will definitely be around 10 pages in the ICLR template.

---

### Official Review · AnonReviewer1 · 2020-10-29
**Lack novelty; overstatement; format violation**

**Rating:** 3
**Confidence:** 5

**Review:**

This paper studies the correlation between the flatness of the converged local minimum and the margin. The authors report experimental results that verify the positive correlation. They suggest using margin-based measures to assess the generalizability. Also, the authors argue that large-batch optimization does not have enough time to maximize margins and hence generalize worse and suggest using it to replace the “misleading folklore” that small-batch methods generalize better because they are able to escape sharp minima. In addition, the authors significantly narrowed the margin which would have violated the policy: “Tweaking the style files may be grounds for rejection.”

Overall, I vote for rejection. The experiments are described in detail and seem correct. However, I was worried that (1) the reported results are not new; and (2) the authors argue existing results are misleading but did not give enough establishment to support their argument. Extraordinary claims require extraordinary evidence. It would be good if the authors can address thesis concerns in the rebuttal session.

Pros:

+ The authors conduct experiments which verify a positive correlation between the margin and the flatness.

Cons:

- The results are not new. It is well-known that (1) margin is a good measure for assessing the generalizability [1-4]; and (2) flatness has a strong correlation with the generalizability as the authors have stated. Combining (1) and (2), it is not surprising margin and flatness has a strong correlation.

- The authors argue that large-batch optimization does not have enough time to maximize margins and hence generalize worse. This argument lacks evidence from either theoretical or empirical aspect.

- The authors argue that it is a misleading folklore that small-batch methods generalize better because they are able to escape sharp minima, still without evidence. In contrast, this has been established in many works; e.g., Sagun et al. (2017) as the authors mentioned.

- The authors significantly narrowed the margin. As stated in the template: “Tweaking the style files may be grounds for rejection.”

Questions: It would be good if the authors can address the cons.

[1] Vladimir Vapnik and Vlamimir Vapnik. Statistical learning theory. Wiley New York, 1:624, 1998.
[2] Peter Bartlett and John Shawe-Taylor. “Generalization performance of support vector machines and other pattern classifiers.” In Advances in Kernel Methods: Support Vector Learning, pages 43–54, 1999.
[3] Vladimir Koltchinskii, Dmitry Panchenko. “Empirical margin distributions and bounding the generalization error of combined classifiers.” The Annals of Statistics, 30(1):1–50, 2002.
[4] Ben Taskar, Carlos Guestrin, and Daphne Koller. “Max-margin Markov networks.” In Advances in Neural Information Processing Systems, pages 25–32, 2004.

---

### Author Response · Authors · 2020-11-10
**Rebuttal**

Dear Reviewers,

Thank you for your feedback. We will present our rebuttal within the next days.

As requested, we have uploaded our paper such that it conforms to the ICLR style guide. We have refrained from making any major textual edits. To fit the content within the page limit, we adjusted the images to the correct margins and lightly edited some captions.

We sincerely apologize for any inconvenience this has caused.

Sincerely, Authors.

---

> ### Author Response · Authors · 2020-11-24
> **No Rebuttal**
>
> Dear Reviewers,
>
> We want to thank you once more for the helpful feedback.
>
> Our work was meant to explore the relationship between robustness and flat minima further. Hence, the first contribution was meant as the core part of the paper and the remainder as an application of that insight. We now believe that exploring how classification margins and flatness relate using the mean squared error and cross-entropy loss will be better. While we believe that this approach has merit, we were unable to make satisfactory changes without altering the paper too much.
>
> We will incorporate your feedback in the next iteration of this work.
>
> Best, Authors

---

### Decision · Program_Chairs · 2021-01-07
**Final Decision**

**Decision:**

Reject

**Comment:**

All reviewers explain in detail, why they think the paper should not be accepted. Besides fixing an initially criticized format violation, the authors did not respond to any of the concerns raised the reviewers, and in fact, they partially agree that more work in another direction needs to be done.